# Physical Mechanism of Nonlinear Spectra in Triangene

**DOI:** 10.3390/molecules28093744

**Published:** 2023-04-26

**Authors:** Na Zhang, Weijian Feng, Hanbo Wen, Naixing Feng, Hao Sheng, Zhixiang Huang, Jingang Wang

**Affiliations:** 1Liaoning Provincial Key Laboratory of Novel Micro-Nano Functional Materials, College of Science, Liaoning Petrochemical University, Fushun 113001, China; zhang_na01@126.com (N.Z.); ffweijian@163.com (W.F.); 13470600357@163.com (H.W.); jingang_wang@lnpu.edu.cn (J.W.); 2The Key Laboratory of Intelligent Computing and Signal Processing, Ministry of Education, Anhui University, Hefei 230601, China; zxhuang@ahu.edu.cn; 3Anhui Province Key Laboratory of Target Recognition and Feature Extraction, Lu’an 230601, China; 4The Information Materials and Intelligent Sensing Laboratory of Anhui Province, Anhui University, Hefei 230601, China; 5The Key Laboratory of Electromagnetic Environmental Sensing of Anhui Higher Education Institutes, Anhui University, Hefei 230601, China

**Keywords:** OPA, TPA, ECD, charge transfer, triangulene

## Abstract

In this work, we theoretically investigate the linear and nonlinear optical absorption properties of open triangulene spin chains and cyclic triangulene spin chains in relation to their lengths and shapes. The physical mechanism of local excitation within the triangular alkene unit and the weak charge transfer between the units are discussed. The uniformly distributed electrostatic potential allows the system to have a small permanent dipole moment that blocks the electronic transition in the light excitation such that the electronic transition can only be carried out between adjacent carbon atoms. The one-photon absorption (OPA) spectra and two-photon absorption (TPA) spectra are red-shifted with the addition of triangulene units compared to N = 3TSCs (triangulene spin chains, TSCs). Here, TPA is mainly caused by the first step of the transition. The length of the spin chain has a significant adjustment effect on the photon cross-section. TSCs of different lengths and shapes can control chirality by adjusting the distribution of the electric dipole moment and transition magnetic dipole moment. These analyses reveal the photophysical properties of triangulene and provide a theoretical basis for studying the photophysical properties of triangulene and its derivatives.

## 1. Introduction

One-dimensional carbon nanostructures, such as carbon nanowires (NWs), carbon nanorods, and carbon nanotubes (NTs), exhibit excellent physical and chemical properties due to their unique microstructures [1,2]. These systems have two limited dimensions, providing an opportunity to study the dependence of the optical properties of electron transport on dimensional constraints [3,4]. Since its successful isolation, graphene has been shown to offer exceptional optical absorption properties and extremely nonlinear optical properties due to its interstitial-free linear electronic structure, including two-photon absorption (TPA), optical limiting, and strong optical anisotropy in the ultraviolet frequency band [5].

In the field of carbon nanomaterials, methods for obtaining new properties often directly alter the physicochemical properties of the material by means of doping, electric fields, functionalization, etc. The disadvantage of this method is that there are more influencing factors that are not easily controlled [6,7,8]. Another method is to accurately construct graphene nanostructures with specific configurations at the atomic level. Conformations include curling, cropping, and stacking, thereby changing the boundary arrangement, rotation angle, and curl angle, which will form nanostructures with different atomic configurations [9]. For example, after adding the magic angle stacking coupling, the interlayer interaction between the two dislocation layers of the twisted bilayer graphene significantly changes the low-energy band structure. This change creates new electronic characteristics different from those of inherent graphene, and its spectral characteristics can identify the rotation angle between the two layers [10,11].

Unlike the two-dimensional stacking method, another triangular graphene segment with regular serrated edges is considered to have different magnetism than graphene because of its unique characteristics of one or more unpaired delocalized electrons [12]. Triangulene contains bi-radical polycyclic aromatic molecules (nano-graphene) consisting of six benzenoid rings. Studies have shown that the key to determining the electronic and magnetic properties of triangulene is size and shape. The physical and chemical properties of triangulene are dependent on the corresponding molecular topology [13,14,15]. Triangulene is a planar structural unit that allows multiple units to form spin-chain polymers with conjugated bonds in a solid state [16]. In 2021, Shantanu Mishra’s team fabricated one-dimensional spin chains containing S = 1 polycyclic aromatic hydrocarbon triangulene as building blocks via surface synthesis techniques [17] and detected magnetic excitation related to the length of open and ring spin chains at the atomic scale in addition to probing the length-dependent magnetic excitation of open spin chains and ring spin chains at the atomic scale. The magnetic properties of triangulene spin chains (TSCs) have been extensively studied both theoretically and experimentally and are expected to provide an ideal platform for exploring the spin physics of S = −1 chains [18]. However, this material’s linear and nonlinear light absorption properties have not been fully theoretically investigated.

The molecular structure can be used to control the optical properties. For example, by changing the length and link mode of the conjugated chain of triangulene, the electronic coupling strength between units can be controlled [19]. Optical properties will change significantly as the length of the triangulene spin chain (depending on the number of units) and the way it is connected change. Surface synthesis to prepare extended triangulate-based nanostructures and restrict chain growth [20,21] produced short open TSCs of different lengths (N = 2, 3, 4, 6, 8 TSCs) and cyclic triangulene spin chains (cTSCs; N = 5, 6, 16 cTSCs), where N represents the number of triangulene units in the spin chain (Figure 1). In one-photon absorption (OPA), the electronic transition characteristics are described by a linear combination of the excited state wave function and the single excited configuration function [22]. TPA is a third-order nonlinear process. This process involves a two-step transition, requiring the absorption of two photons to transition from a lower to a higher energy level, which significantly reduces the transition probability, whereas OPA requires the absorption of only one-photon. This method is based on the sum-of-states (SOS) theory to visualize the two-step transition process [23], and the TPA spectrum calculated by the SOS-based TPA analysis program agrees well with the experimental results [24]. We calculated the circular dichroism of the spin chain of triangulene and visualized the generalized chirality of the system by analyzing the electric dipole moment and magnetic dipole moment of the electromagnetic wave interaction in the system [25].

## 2. Results and Discussion

### 2.1. OPA Spectrum

Triangulene consists of six aromatic benzene ring-like units with radial p-orbitals. The presence of two unpaired electrons [26] exists in a single triangulene. Under optical excitation, the two unpaired electrons form electron–hole pairs, which determine the optical absorption properties of triangulene; these pairs have a spin-chain structure containing multiple unpaired electrons. Some relevant information is given in Appendix A. The spin density distribution is shown at the edges of the serration [26] (Appendix A). This distribution can hinder or prevent the compounding of photoexcited electron and hole pairs in the catalyst and the promotion of photogeneration for electron–hole pairs [27].

In order to investigate the effect of conformational changes on the optical absorption properties of TSCs, we selected several TSCs that were representative of conformation (Figure 1). On the basis of the TSCs (N = 2TSCs) connected by two triangulene units, the TSCs units were added, and the triangulene connection sites were changed. We then constructed 10 kinds of TSC quantum dots with different structures and periodic micro-nano sizes. Each two adjacent triangulene units were joined in such a way that the C-C bond centers at the junction points were combined as an axis of symmetry. The corresponding OPA spectra are presented in Appendix A. The vertical coordinate is the molar absorption coefficient, the magnitude of which can be used to compare the probability of electrons transitioning between the energy levels of different compounds. Their absorption peaks are located in the UV region between 200 and 400 nm and consist mainly of two strong absorption peaks. The primary absorption peaks of N = 2, 3, 4-1, 4-2, 5c, 6, 6cTSCs were caused by S_73_, S_66_, S_62_, S_62_, S_91_, S_94_, and S_98_, and the secondary absorption peaks were caused by S_38_, S_37_, S_36_, S_35_, S_55_, S_53_, and S_55_, respectively. Since N = 3TSCs also has an excited state with a similar vibronic intensity near S_66_, when the intensity of an oscillator is broadened into the curve of the absorption spectrum, it is subjected to the intensity of that oscillator and forms an even stronger absorption peak. Thus, although S_37_ has a higher oscillator intensity, it has a smaller molar absorptivity. With the addition of a triangulene unit, the two main absorption peaks of N = 4-1TSCs were red-shifted. The main absorption peak was shifted to 261.1 nm, caused by S_62_, and the secondary absorption peak was caused by S_36_ near 297 nm (Appendix A). Appendix A shows the OPA spectrum of 4-2TSCs with a main absorption peak of $\;\lambda_{max}$ = 261.4 nm, caused by S_62_. The secondary absorption peak was caused by S_35_ at 297.1 nm, which is spectroscopically very close to N = 4-1TSCs. This small gap may have occurred because triangulene forms two completely different conformations due to different attachment sites, resulting in different excitation energy between the two configurations at the same wavelength. This difference leads to an increase with increasing size, indicating that the change in conformation has a significant effect on the optical absorption properties of triangulene. When the TSCs form a closed six-membered ring, $\lambda_{max}$ = 257.6 nm of the main absorption peak of the N = 6c TSCs is caused by the excited state S_98_, and the secondary absorption peak is caused by S55 near 294.5 nm (Appendix A). The distinctive feature of the absorption spectrum of N = 6cTSCs is greatly enhanced absorbance. The main absorption peaks show a color enhancement effect, mainly due to the large vibrational intensity of the system.

Figure 2b shows the TPA spectrum of triangulene. The vertical coordinate is the TPA cross-section. TPA is a third-order nonlinear optical absorption process caused by the absorption of two photons at twice the excitation wavelength of OPA. The main absorption peak of the TPA spectrum is located at 600–800 nm. Therefore, compared to the OPA spectrum, the main site is excited by high-energy-region photoluminescence, which can effectively avoid optical damage to the material caused by the excitation beam. For TPA, attention needs to be paid to excited states with large TPA cross sections. Appendix A shows the TPA spectra of four triangulene spin chains with large two-photon cross sections. After changing the attachment point of triangulene, $\lambda_{max}$ = 544.5 nm for N = 3TSCs, where the absorption peak is mainly caused by S_50_; $\lambda_{max}$ = 573 nm for N = 4-1TSCs, where the absorption peak is mainly caused by S_40_; and $\lambda_{max}$ = 573.6 nm for N = 4-2TSCs, with absorption peaks mainly caused by S_40_ and reduced two-photon cross sections compared to N = 4-1TSCs. After changing the connection point of triangulene, N = 4-2TSCs and $\lambda_{max}$ = 573.6 nm. Here, the absorption peak is mainly caused by S_40_, and the two-photon cross-section decreases compared to N = 4-1TSCs; N = 6cTSCs and $\lambda_{max}$ = 573.8 nm are mainly caused by S_60_. Appendix A presents a composite diagram of the two-step transition summation terms of the four configurations of the TPA spectra. It can be seen that the molar absorption intensity of TPA with N = 3TSCs is the largest. In addition, the TPA peak of N = 6cTSCs changes from two to one, mainly because the excited state of the two-photon cross-section is concentrated around 550 nm.

In order to study the physical mechanism of the change in absorption spectrum caused by the change of the triangulene spin chain configuration in essence, it is necessary to analyze the transition density matrix (TDM) of excited states. For a single electronic state, the density matrix is a disguised form of the wave function of a specific electronic state. TDM can reflect the transition between two states, and the electronic distribution from the ground state to the excited state can be easily obtained, enabling one to determine and analyze the detailed optical absorption characteristics of the triangulene spin chain. The real spatial form of TDM between the ground state and excited state in a multi-electron system is as follows:(1)$$T(r;r^{\prime })\equiv T(r_{1};r_{1}^{\prime })=\int \Phi^{0}(x_{1},x_{2},\cdots x_{N})\Psi^{exc}(x_{1}^{\prime },x_{2},\cdots x_{N})d\sigma_{1}dx_{x}dx_{3}\cdots dx_{N}$$
where $\Phi^{0}$ is the ground state wave function, $\Psi^{exc}$ is an excited state wave function, x is the spin + space coordinate of the electron, σ is the spin coordinate, and r is the space coordinate. When calculating the excited state using TDDFT with the reference method, the excited state wave function is described by the linear combination of various single excited configuration functions, and TDM is expressed as
(2)$$T(r;r^{\prime })=\sum_{i}\sum_{a}w_{i}^{a}\varphi_{i}(r)\varphi_{a}(r^{\prime })$$
where TDM is a six-dimensional function. The, take the diagonal elements of Formula (2) to obtain the three-dimensional transition density T(r):(3)$$T(r)=\sum_{i}\sum_{a}w_{i}^{a}\varphi_{i}(r)\varphi_{a}(r)$$
where, the transition of orbital electrons from occupied orbit i to empty orbit a can be easily observed through image processing. The larger the T(r) is, the more obvious the overlap of the i and a orbits becomes. Generally, electron excitation includes the excitation of multiple orbitals, so it is broadly described as “hole → electron”. The definition of a hole is as follows:(4)$$\rho^{\mathrm{hole}}(r)=\rho_{\left(loc\right)}^{\mathrm{hole}}(r)+\rho_{\left(cross\right)}^{\mathrm{hole}}(r)$$
which is composed of the local term $\rho_{\left(loc\right)}^{\mathrm{hole}}(r)={\sum_{i\to a}\left(w_{i}^{a}\right)}^{2}\varphi_{i}\varphi_{i}-{\sum_{i\leftarrow a}\left(w_{i}^{a}\right)}^{2}\varphi_{i}\varphi_{i}$ and the $\rho_{\left(cross\right)}^{\mathrm{hole}}(r)=\sum_{i\to a}\sum_{j\neq i\to a}w_{i}^{a}w_{j}^{a}\varphi_{i}\varphi_{j}-\sum_{i\leftarrow a}\sum_{j\neq i\leftarrow a}w_{i}^{\prime a}w_{j}^{\prime a}\varphi_{i}\varphi_{j}$ cross term, where r is the coordinate vector, φ is the orbital wave function, i or j is the occupied orbital label, a or b is the empty orbital label, and the definition of the electron is similar. When the absolute value of the product of the wave functions of the two orbits is large, the greater the transition density is, the greater the overlap of the corresponding i and a orbits becomes. Conversely, the smaller the absolute value of the product of the two orbital wave functions is, the smaller the overlap of i and a becomes. Therefore, it can be approximately considered that the place with high transition density will present a large overlap of holes and electrons.

Figure 3 shows the TDM and hole-electron pair density of the secondary absorption peak of the triangulene spin chain with different configurations. TDM is displayed in the form of a heat map, and the abscissa and ordinates are atomic numbers. The brighter the matrix element color is, the greater the transition density is. Figure 3a shows the TDM and hole-electron pair density of N = 2TSCs in S_38_. TDM shows that there are significant green and yellow areas on the main diagonal. There are also some transition density distributions on both sides of the main diagonal, which are caused by the charge transfer. The density of the hole-electron pairs shows that the holes are mainly distributed at the junctions of two triangulene sheets, and the electrons are distributed on both sides. Therefore, the electronic transition of S_38_ has obvious charge transfer characteristics. After adding a triangle, TDM shows that the transition density is almost completely distributed on the main diagonal, corresponding to atoms 25–40, namely, the triangle unit in the middle of the triangulene dimer. The blue isosurface on the hole-electron pair represents the area where the electron decreases during the electron transition, and the red isosurface represents the area where the electron increases. In addition to a small number of hole-electron pairs on the triangulene units on both sides, the main isosurface is distributed on the triangulene in the middle, which is very consistent with the distribution of transition density, as shown in Figure 3b. In addition to the relatively high transition density of the intermediate triangulene, there are two areas with weak brightness at the upper-left and upper-right sides of the TDM, which represent the transition density on the left and right sides of the triangulene trimer, respectively. Because these transition densities are mainly distributed on the diagonal, the TDM reflects the local excitation characteristics between adjacent carbon atoms. After adding a triangulene unit, the main diagonal of TDM shows two connected areas with similar brightness, which can be speculated to represent the local excitation of the middle two pieces of triangulene in N = 4-1TSCs. The hole-electron pair density diagram on the right also indicates that the hole-electron is mainly distributed at the connection point of the middle two pieces of triangulene, as shown in Figure 3c. After changing the connection point of triangulene, the distribution of bright areas on TDM is similar, belonging to the local excitation of the middle two pieces of triangulene, as shown in Figure 3d. The ring triangulene spin chain composed of five triangulene N = 5cTSCs also belongs to local excitation in S_55_, and the hole-electrons are mainly distributed at the junction of triangulene units on both sides (Figure 3e). After triangulene forms a closed six-member ring, TDM shows that the transition density is also divided into two bright regions, but the transition density of this system is the smallest of the four configurations. From the perspective of the density of hole-electron pairs, the distribution of the hole-electron-equivalent surface is central symmetry. At the same isosurface value (iso value = 0.001), the density of the hole-electron pair is the smallest (Figure 3f), which shows that the electron transition is weakened in this stable ring structure.

Compared to the secondary absorption peak, the main absorption peak has higher oscillator strength. This result shows that the absorption intensity of the system is enhanced. Figure 4 shows the TDM and hole-electron pair densities of the excited states of the main absorption peak of the triangulene spin chain with different configurations. The TDM of N = 2TSCs shows that the excitation characteristics of S_73_ belong to local excitation and the hole-electrons overlap at the junction of two triangulene units, as shown in Figure 4a. The excitation characteristics of N = 3TSCs in S_66_ are opposite to the secondary absorption peak. From the TDM, the bright yellow and green areas are mainly located at the lower left and upper right areas of the main diagonal. Compared to S_37_, the distribution range of the bright regions is greatly enlarged, and their atomic numbers correspond to triangulene on both sides of the system. From the perspective of hole-electron pair density, the hole-electron isosurface is very significant and almost completely distributed on both sides of triangulene, belonging to local excitation between adjacent carbon atoms of triangulene on both sides, as shown in Figure 4b. The electron excitation of N = 4-1TSCs in S_62_ was transferred to the middle two triangulene units. TDM shows that compared to S_36_, the degree of electronic excitation is enhanced. Based on the density of the hole-electron pairs, there is also electron distribution on the middle two triangulene junction points, as shown in Figure 4c. After changing the connection point, the electronic excitation characteristics of N = 4-2TSCs in S_62_ also belong to local excitation, as shown in Figure 4d. The electronic transition of N = 5cTSCs in S_91_ comes from the triangulene unit in the middle part, which belongs to local excitation, as shown in Figure 4e. The degree of electronic excitation in the closed six-member ring in S_98_ is still the smallest, belonging to weak local excitation. Compared to the secondary absorption peak, the hole-electron pairs are distributed on the triangulene unit adjacent to the two directions, as shown in Figure 4f. Figure 5 shows the TDM and hole-electron pair density of the main absorption peak of the N = 6, 8-1, 8-2, and 16cTSCs, where the N = 8-1, 8-2, and 16cTSCs have only one absorption peak. Compared to the previous structure, the transition density of TDM is only distributed on the main diagonal as the system increases. Here, the isosurface of the hole-electron is significantly reduced, indicating that the degree of electron transition is weakened (Figure 5a–c). When the quantum dot is composed of 16 triangulene units, the distribution of the hole-electron disappears, as shown in Figure 5d.

### 2.2. Analysis of Transition Index of Excited State

Visualization of the TDM and hole-electron pair density is used to qualitatively investigate the electronic transition characteristics of excited states. In order to investigate the characteristics of electron excitation, the wave function analysis of electronic states is also needed from a quantitative perspective. We also calculated the transition index of one-photon excited states and selected four triangulene spin chain structures, as shown in Table 1. The *H* index represents the overall distribution breadth of hole-electron pairs, which is defined as
(5)$$H^{}\;\mathrm{index}=(|\sigma_{\mathrm{ele}}|+|\sigma_{\mathrm{hole}}|)/2$$
where $\sigma_{\mathrm{ele}}$ and $\sigma_{\mathrm{hole}}$ are used to characterize the spatial distribution breadth of electrons and holes, including the root mean square difference of the distribution of electrons and holes in the three directions of xyz. x in the direction of $\sigma_{\mathrm{hole}}$ is defined as
(6)$$\sigma_{\mathrm{hole},x}=\sqrt{\int {\left(x-X_{\mathrm{hole}}\right)}^{2}\rho^{\mathrm{hole}}(r)dr}.$$

It can be seen that the *H* index of the excited state at the secondary absorption peak increases with size, which is caused by a gradual increase in the system structure. In addition, the configuration change will cause a change in excited state energy, with the excitation energy of S_37_ being the largest among the four configurations. The D index is the distance between the hole and the electron centroid, which is derived from the Cartesian coordinates of the hole and the electron centroid and obtained from the following formula:(7)$$D^{}\;\;\mathrm{index}=\sqrt{{\left(D_{x}\right)}^{2}+{\left(D_{y}\right)}^{2}+{\left(D_{z}\right)}^{2}}.$$

The *D* index of N = 3TSCs in S_66_ reached 0.173 Å, which is the largest among all one-photon excited states. This result was mainly due to the location of the hole-electron exchange on both sides of the triangulene. Additionally, on a triangulene, the holes are mainly distributed at the edge of the triangulene, while the electrons are mainly distributed in the interior. Therefore, compared to other excited states, the electron migration path in S_66_ is longer. Corresponding to the distance between the centroid of the hole-electron is the separation degree t of the electron and the hole, which is defined as
(8)$$t^{}\;\mathrm{index}=D^{}\;\mathrm{index}-H_{CT}$$
where $H_{CT}$ is the average extent of electrons and holes in the direction of charge transfer, $H_{CT}=|H-\mu_{CT}|$ and H are the vector sum of the average extent of holes in the three directions of xyz, and $\mu_{CT}$ is the unit vector in the direction of charge transfer. Generally, a combination of these two indexes can be used to examine the strength of charge transfer in the process of electron excitation. Table 1 and Table 2 show that all excited states *t* < 0, indicating that there is no significant separation of holes–electrons in the direction of charge transfer because the distance between the center of mass of the hole and the electron is far less than their average extension on the triangulene spin chain. We also define the $S_{r}$ index, which is used to characterize the coincidence degree of electrons and holes. The physical meaning of this index is opposite to that of the t index. The definition is as follows:(9)$$S_{r}{}^{}\;\mathrm{index}=\int S_{r}(r)dr\equiv \int \sqrt{\rho^{\mathrm{hole}}(r)\rho^{\mathrm{ele}}(r)}.$$

This index is the full space integral of the $S_{r}$ function $S_{r}(r)=\sqrt{\rho^{\mathrm{hole}}(r)\rho^{ele}(r)}$. When the value is 1, the electron and hole completely coincide. Based on the table, the Sr indexes of all excited states are very close to the maximum value of 1, indicating that the overlap of electrons and holes is quite large. The above analysis of the transition index of the electronically excited states shows that local excitation between adjacent carbon atoms dominates the electronic transition of the triangulene spin chain quantum dots. After adding triangulene, with the growth of the conjugated chain, the energy difference between the levels of the triangulene spin chain decreases. This decrease reduces the excitation energy of electrons and causes absorption to move toward the long wave spectrum.

### 2.3. Molecular van der Waals Surface Electrostatic Potential

The main absorption peak of the triangulene spin chain shows the characteristics of local excitation after absorbing photons. The molecular dipole moment is the key factor used to determine the oscillator strength in photo-induced electron transition. In order to analyze the influence of the electrostatic interaction of the triangulene spin chain on the transition dipole moment, the molecular van der Waals surface electrostatic potential mapping of triangulene in different configurations, and the corresponding extreme values are plotted, as shown in Figure 6 (the red-equivalent surface is the area with positive electrostatic potential distribution, the blue-equivalent surface is the area with negative electrostatic potential distribution, and the red and blue spheres represent the maximum and minimum electrostatic potential, respectively), as shown in Figure 6. It can be seen that the electrostatic potential distribution of the triangulene trimer spin chain has a large negative value. This value reflects the large pi conjugation between adjacent carbon atoms in triangulene as a whole, and the edge electrons of H modification show positive values due to the covalent bond close to the carbon atom. The maximum point of electrostatic potential is mainly distributed in the area where the two pieces of triangulene are connected at the edge (about 21.61 kcal/mol). The minimum point of electrostatic potential (about −14.1 kcal/mol) is mainly concentrated near the fixed point of triangulene, as shown in Figure 6a, because the covalent bond formed by the overlapping of the p orbitals of the edge carbon atoms and the s orbitals of the hydrogen atoms is lower than the overlapping energy of the p orbitals of the middle carbon atoms and the three nearby carbon atoms. The electrostatic potential distribution on the van der Waals surfaces of the other three configurations is similar. As shown in Figure 6b–d, this uniformly distributed electrostatic potential has a small permanent dipole moment, produces a larger oscillator strength, and increases the transition probability between excited states. However, due to the large pi-conjugated structure in triangulene, which hinders the degree of electronic transition in light excitation, the electronic transition can only be carried out between adjacent carbon atoms, resulting in strong local excitation characteristics within the system.

### 2.4. Two-Photon Transition

Photon-excited spectral analysis shows that the absorption of the triangulene spin chain is located in the ultraviolet region, which limits its application. Here, the large-section TPA with two photons at the same time under an intense laser pulse presented many excellent characteristics, such as the ability to produce an excited state with half the nominal excitation energy, improve the penetration ability of the absorbing or scattering medium, and transfer the optical absorption to the visible light region, greatly improving the application ability. Therefore, we calculated the first and second transition characteristics for the TPA of the triangulene spin chain through the three-state model [1,2] of Formula (1). Table 3 shows the transition dipole moment elements of the maximum two-photon cross-section. Here, N = 3TSCs have the largest TPA cross section among the four configurations, and the maximum value is caused by the two-step transition of S_50_. Figure 7a,b show the TDM and hole-electron pair density from the ground state to the excited state ($S_{0}\to S_{2}$). The TDM shows that the transition density is mainly distributed in the middle of the diagonal, and the hole-electron density value is concentrated on the intermediate triangulene. Therefore, the first step of transition belongs to strong local excitation between adjacent carbon atoms on the intermediate triangulene. The second transition comes from the second part of Formula (1), that is, the electronic transition from the excited state to the final state ($S_{2}\to S_{50}$). TDM shows that in addition to the transition density on the main diagonal, there are weak bright areas on the non-diagonal, as shown in Figure 7c. Combined with the density of hole-electron pairs, it can be seen that the electrons transfer from the left triangulene of N = 3TSCs to the middle triangulene, as shown in Figure 7d. Therefore, the second transition can be observed as local charge transfer excitation.

As shown in Table 3, after adding one triangulene, the maximum two-photon cross section of N = 4-1TSCs has an S_40_ contribution, and its two-photon transition consists of two channels, where the intermediate state of channel I is S_36_. From the perspective of TDM and hole-electron pair density, the first step of transition $S_{0}\to S_{36}$ belongs to local excitation between the two adjacent atoms of the middle triangle alkene, as shown in Figure 8a,b. The transition density of transition $S_{36}\to S_{40}$ in the second step is significantly lower than that in the first step, and there is almost no hole-electron pair density (Figure 8c,d). When the intermediate state is S_36_, the TPA is mainly caused by the first step. The intermediate state of the other channel in the two-photon transition of S_40_ is S_39_. The TDM of $S_{0}\to S_{39}$ shows that the first step of the transition still belongs to local excitation. Compared to channel I, the degree of local excitation is enhanced. Based on the distribution of the hole-electron pair density, the hole-electron distribution has a very obvious distribution on the four triangles (Figure 8e,f), and the transition characteristics of the second step of transition $S_{39}\to S_{40}$ are similar to those of channel I. The two-photon transition of N = 4-1TSCs shows that the TPA of the triangulene spin chain is mainly caused by the first transition after the system increases.

It can be seen from the TPA spectrum in Figure 2 that after changing the triangulene junction point, the absorption spectra of two triangulene spin chains with the same size are very similar. Table 3 shows that the maximum two-photon cross section of N = 4-2TSCs is caused by S_40_ and consists of two channels. The dipole moment distance of the first step is smaller than that of the second step. The intermediate state of channel I is S_35_, and the TDM and hole-electron pair density of the two-step transition is very similar to those of N = 4-1TSCs, as shown in Figure 9a–d. The other channel, the intermediate state, is S_38_. The TDM of $S_{0}\to S_{38}$ shows that in addition to the relatively large transition density on the diagonal, there is also significant transition density distribution in the non-diagonal region, as shown in Figure 9e. Based on the density of hole-electron pairs, the electrons are mainly concentrated on the two triangulene units in the upper half, as shown in Figure 9f. This result indicates that the electron transfer direction extends from the two triangulene units in the lower part to the adjacent triangulene units in the upper part, presenting the characteristics of charge transfer. The second transition $S_{38}\to S_{40}$ belongs to weak charge transfer excitation, as shown in Figure 9g,h. Therefore, a comparison of the two configurations shows that the charge transfer ratio of the two-step transition can be significantly increased by changing the triangulene junction point, and the electronic transition ability can be enhanced.

After triangulene forms a closed six-member ring, the maximum two-photon cross-section is caused by S_61_. The two-photon transition process includes two channels. The intermediate state of channel I is S_56_. From the perspective of TDM, the first step of transition $S_{0}\to S_{56}$ belongs to local excitation. As shown in Figure 10a, the density of the hole-electron pairs shows that the hole-electrons are distributed on the adjacent triangulene units on the upper and lower sides (Figure 10b). Combined with the second transition characteristics of the triangulene spin chain in the previous three configurations, we found that with a further increase in size, the electronic transition degree of the second transition of N = 6cTSCs further decreased, and the distribution of hole-electrons almost disappeared (Figure 10c,d). The intermediate state of channel II is S_55_, and the first step of transition $S_{0}\to S_{55}$ also belongs to local excitation. The second step of transition $S_{55}\to S_{61}$ has the same excitation characteristics as channel I. However, unlike channel I, the hole-electron distribution is exactly the opposite, with hole-electrons mainly distributed on left and right single triangulene units.

### 2.5. Electron Circular Dichroism

Triangulene can not only significantly change the linear optical absorption characteristics and nonlinear optics of TSCs by adjusting the size and connection mode to form different configurations but also has a significant impact on the corresponding chiral properties, as shown in Appendix A. As the size increases, the circular dichroism intensity of N = 3TSCs increases, the circular dichroism intensity of N = 4-1TSCs becomes very weak, and the circular dichroism intensity of N = 8-1TSCs is significantly enhanced, as shown in Appendix A. In addition, under the same size, circular dichroism becomes significantly different due to the use of different connection methods, as shown in Appendix A. Circular dichroism is an important method used to characterize the chirality of a system and is derived from the asymmetric response of the electromagnetic interaction between the system and the electromagnetic wave to the transition electric dipole distance and the transition magnetic dipole moment. Figure 11a shows the ECD spectrum of N = 3TSCs at 280–350 nm (the left longitudinal axis represents the differential absorption of the system to the left and right rotatory light, and the right longitudinal axis is the rotor intensity), which is composed of a positive absorption peak and a negative absorption peak. The positive absorption peak $\lambda_{max}$ is mainly caused by S_30_ at 301.8 nm, and the negative absorption peak $\lambda_{max}$ is caused by S_26_ at 321.4 nm. After adding a triangulene unit, the circular dichroism of N = 4-1TSCs almost disappears from the perspective of the absorption intensity of the ordinate, which is due to the formation of a triangulene spin chain with a symmetrical structure after adding a triangulene unit. Compared to N = 3TSCs, the molar absorption intensity of N = 4-1TSCs is positive, and the main and secondary absorption peaks near the same wavelength are, respectively, caused by S_36_ at 297.0 nm and S_21_ at 322.3 nm, as shown in Figure 11b. After changing the linkage sites, the changing trend of ECD of N = 4-2TSCs is similar to that of N = 4-1TSCs, but the intensity of the molar absorption peak is greatly enhanced. The $\lambda_{max}$ of the main absorption peak and the secondary absorption peak are, respectively, caused by S_35_ at 288.7 nm and S_21_ at 322.4 nm, as shown in Figure 11c. When triangulene forms a closed six-member ring, the intensity of the absorption peak decreases significantly. Compared to N = 3TSCs, the ECD absorption peak of the same wavelength N = 6cTSCs is reversed, and the $\lambda_{max}$ of the negative absorption peak is caused by S_33_ at 322.3 nm, as shown in Figure 11d.

### 2.6. Chiral Physical Mechanism

In order to deeply analyze the physical mechanism of ECD chirality inversion after the configuration changes of TSCs, the visual transition electric dipole moment density (TEDM, where the positive value is purple, and the negative value is yellow) and transition magnetic dipole moment density (TMDM, where the positive value is blue, and the negative value is yellow) are drawn, and the isosurface value is set to 0.008. At 321.4 nm, N = 3TSCs have a strong transition dipole moment in the X component, mainly distributed on the triangulene sheets on both sides. There is almost no transition dipole moment density on the Y and Z components, as shown in Figure 12a–c. The transition magnetic dipole moment is mainly distributed on the Y and Z components, with a large isosurface; these components are clustered on the triangulene sheets on both sides, as shown in Figure 12d,f. After adding a triangulene unit, the distribution of the transition electric dipole moment density and the transition magnetic dipole moment density of the two configuration molecules are the same in the Z component, but there are significant differences in the X and Y components. The isosurface of the transition dipole moment density on the three components is not very large. The transition dipole moment on the Y component spreads to the entire triangulene spin chain, as shown in Figure 12g–i. The distribution of the transition magnetic dipole moment density is also similar, and the transition magnetic dipole moment on the X component also presents a diffusion distribution trend, as shown in Figure 12j–l. Therefore, overall, the transition electric dipole moment and transition magnetic dipole moment of N = 4-1TSCs are less than those of N = 3TSCs. The transition dipole distance density of N = 4-1TSCs on the X component is mainly distributed on the middle two triangulene units. However, the transition magnetic dipole moment is mainly distributed on both sides. The Y component is exactly the opposite, with the transition electric dipole moment density distributed on both sides and the transition magnetic dipole moment density distributed in the middle, and the distribution range is complementary. In this case, the transition electric dipole distance and transition magnetic dipole strength are completely opposite in the same direction on N = 3TSCs because TEDM and TMDM can reveal the responses of molecular and electromagnetic waves of triangulene spin chains in the process of light excitation. Because the direction of the electric field and the magnetic field are vertical, the distribution range of these fields is complementary. In addition, from the perspective of rotor strength, ECD shows that the circular dichroic absorption intensity of N = 4-1TSCs became significantly reduced. According to Formula (6) in Section 2.3, the rotor strength depends on the tensor product ${\left|\langle \varphi_{j}|\mu_{e}|\varphi_{i}\rangle \langle \varphi_{j}|\mu_{m}|\varphi_{i}\rangle \right|}^{2}$ of electricity and magnetism in three directions. Because the transition electric dipole moment and transition magnetic dipole moment of N = 4-1TSCs on the X and Y components are small, the overall tensor product becomes smaller; ultimately, the smaller rotor strength can be determined.

Another ECD excited state of N = 3TSCs is S_30_. Compared to S_26_, the transition electric dipole moment density and transition magnetic dipole moment density of S_30_ in the three components are similar, but the distribution trend is very different. The yellow isosurface and purple isosurface of S_26_ on the Y and Z components are relatively independent, so the positive and negative values of the transition magnetic dipole moment are highly separated, as shown in Figure 13a–c. The positive and negative values of the transition magnetic dipole moment density of S_30_ on the Y and Z components are evenly distributed, as shown in Figure 12d–f, which is why circular dichroism changes from negative to positive. After adding a triangulene unit, the transition electric dipole of S_36_ is mainly caused by the Y component, as shown in Figure 13h. The transition magnetic dipole is mainly caused by the Z component, and the X component also contributes a portion of the transition magnetic dipole moment (as shown in Figure 13j–l). Compared to N = 3TSCs, the transition dipole moment of these components is mainly caused by a single square quantity. However, the density of the transition magnetic coupling of S_30_ is larger in the Y and Z components, while the transition magnetic coupling of S_36_ is only larger in the Z component, so the overall tensor product is smaller than S_30_, and its rotor strength is also relatively small. Unlike N = 3TSCs, the absorption peak of S_36_ still shows positive circular dichroism because the isosurface size of the transition electric dipole moment density and the transition magnetic dipole moment density on the three components of S_36_ are similar to those of S_21_, and the distribution trends are also similar. In both, the positive and negative values of the transition electric dipole moment are uniformly distributed on the Y component, and the positive and negative values of the transition magnetic dipole moment are separated on the Z component (as shown in Figure 12g–l and Figure 13g–l. In addition, since the transition dipole moment of S_36_ on the Y component is significantly greater than that of S_21_ (Figure 12h and Figure 13h), it has strong circular dichroism.

After changing the junction site, the ECD absorption peak of N = 4-2TSs at 320 nm is caused by S_23_. The upper part of Figure 14 shows the density of the transition electric dipole moment and transition magnetic dipole moment of S_23_ in Cartesian coordinates. Here, the transition dipole moment is mainly caused by the X component, and there are also some values on the Y component. The transition dipole moment density on the Z component completely disappears, as shown in Figure 14a,b. The transition magnetic dipole moment is mainly caused by the Z component, with only a small distribution on the X and Y components, as shown in Figure 14d–f. Compared to N = 4-1TSCs, the transition magnetic dipole moments of the two systems are similar, and the difference between the system and the photon electromagnetic interaction mainly comes from the transition electric dipole moment. When the connection mode of the added triangulene sheet unit and triangulene trimer is in periodic superposition, the distribution of the transition dipole moment on the three components becomes relatively uniform, and the value is relatively small. When triangulene changes the linkage site and triangulene trimer connection, a large transition dipole moment is generated in the X component. Therefore, this semi-closed triangulene spin chain has a larger tensor product of electricity and magnetism and will present greater rotor strength and stronger circular dichroism.

After triangulene forms a closed six-membered ring structure, the absorption peak near 300 nm is mainly caused by S_33_. Due to its unique ring structure, the complementary characteristics of the transition electric dipole moment and the transition magnetic dipole moment are more significant than the chain structure, and the distribution trend is similar to that of N = 4-2TSCs (Figure 14a–f,g–l.) Therefore, the circular dichroism of the two structures at this wavelength is positive.

The main absorption peak of N = 4-2TSCs is caused by S_35_, whose circular dichroism is significantly enhanced compared to that of S_23_. For the transition dipole moment, the density values on the X and Y components are larger than those of S_23_, as shown in Figure 15a,b. This result mainly shows that the two triangulenes in the upper half of the semi-closed triangulene spin chain also produce a small amount of transition dipole moment density. The change in the transition magnetic dipole moment in the X and Y components is the same. The Z component has an obvious transition magnetic dipole moment density in the whole semi-closed triangulene spin chain, as shown in Figure 15d–f. The large transition electric dipole moment response and transition magnetic dipole moment response on each component make the rotor strength of S_35_ much greater than that of S_23_. The circular dichroism of the closed triangulene six-membered ring N = 6cTSCs at 322.6 nm is negative, and greater chiral inversion occurs compared to N = 4-2TSCs at the same wavelength. The distribution of the transition electric dipole moment density and the transition magnetic dipole moment density in the X and Y components of the two configurations is approximate (Figure 15a–f,g–l). However, there are great differences in the Z component. The transition magnetic coupling of N = 4-2TSCs in the Z component is mainly caused by the yellow isosurface representing a negative value. However, the transition dipole moment density on the closed six-membered ring triangulene spin chain is composed of yellow and purple isosurfaces, and positive and negative values have the same contribution, as shown in Figure 15f–l. This contribution is due to the uneven distribution of transition magnetic dipole moments caused by changes in the triangulene spin chain configuration, resulting in chiral inversion.

## 3. Methods

Quantum chemical calculations for all different configurations of TSCs were performed using the Gaussian 16 software [28]. The geometry was optimized using density functional theory (DFT) [29], the PBE1PBE functional [30], and the def2-SVP [31] basis set combined with DFT-D3 dispersion correction. The Cartesian coordinates of all configurations are listed in the Appendix A. The excited state electron transition characteristics, UV-Vis spectra, and ECD spectra were calculated using the time-dependent density function DFT (TD-DFT) [32], CAM-B3LYP function [33], the def2-SVP basis set, and the dispersion correction method. Based on the SOS method and electron–hole pair density and transition density matrix (TDM), all configuration coefficients, transition electric dipole moment density (TEDM) values, and transition magnetic dipole moment density (TMDM) values for the TPA transition process were determined [34]. The electron–hole pair density was formed by a weighted linear combination of ground, intermediate, and final state wave functions and conformational coefficients. Multiple molecular orbital transitions with corresponding weighting factors were used to represent the TDM, which is the transition between two states through which the dipole moment, the electrostatic potential of this state, is obtained [32]. TEDM and TMDM were calculated from the tangent vectors of the transition density space [35]. The auxiliary calculation and visualization programs used for the drawing process were Multiwfn-3.8 [36] and VMD-1.9.3 [37].

## 4. Conclusions

In this work, we theoretically calculated the linear and nonlinear optical absorption characteristics of triangulene open spin chains (N = nTSCs) and ring spin chains (N = ncTSCs) with different lengths and shapes through DFT theory and the SOS method. With an increase in triangulene units, the UV-Vis absorption spectrum experienced a red shift. TDM and hole-electron pair density analysis showed that the main excitation characteristic of OPA was local excitation in the triangulene unit. The molecular van der Waals surface electrostatic potential showed that this local excitation came from the large pi-conjugated structure inside triangulene, resulting in a small permanent dipole moment generated by the uniform distribution of electrostatic potential, which had an inhibitory effect on electronic transitions. TPA showed that the length and shape of triangulene can adjust the two-photon cross-section. Unlike OPA, intermolecular charge transfer occurred in the first step of the transition. ECD [38] showed that TSCs have chirality, and the visualization analysis of intramolecular electronic interaction showed that the distribution of TEDM and TMDM on Cartesian components will lead to chiral inversion for different sizes and connection modes.

## Figures and Tables

**Figure 1 molecules-28-03744-f001:**
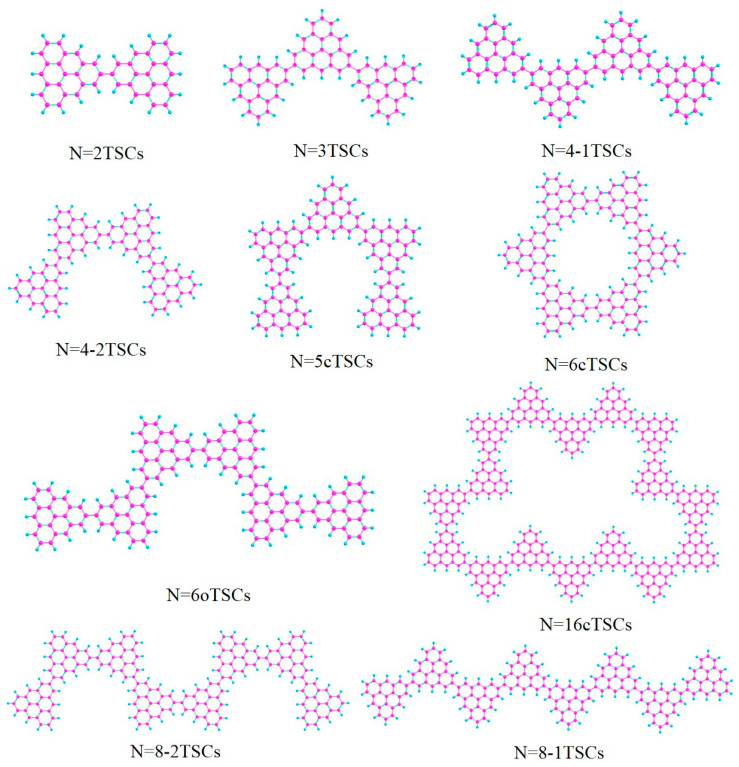
Triangulene spin chains trimers (N = 2, 3, 4-1, 4-2, 6, 8-1, 8-2TSCs) and triangulene cyclic spin chains (N = 5c, 6c, 16cTSCs).

**Figure 2 molecules-28-03744-f002:**
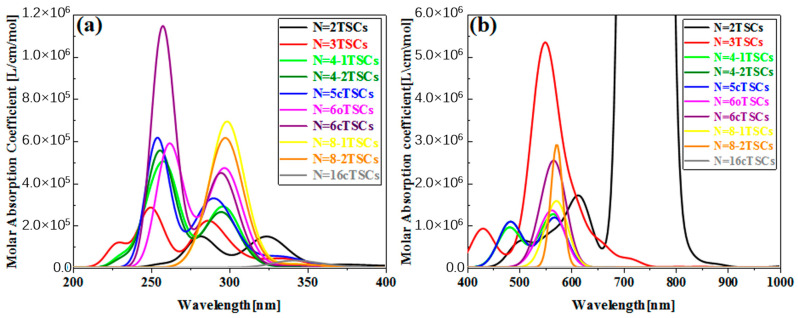
The OPA spectra (**a**) and TPA spectra (**b**) of the triangulene spin chain.

**Figure 3 molecules-28-03744-f003:**
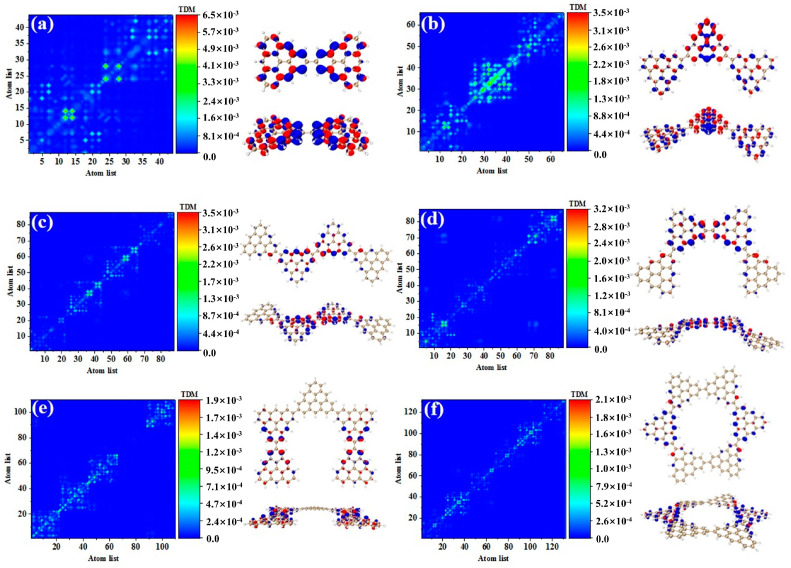
TDM and hole-electron pair density of one-photon excited states of secondary absorption peaks of triangulene spin chain N = 2 (**a**), 3 (**b**), 4-1 (**c**), 4-2 (**d**), 5c (**e**), 6c (**f**) TSCs.

**Figure 4 molecules-28-03744-f004:**
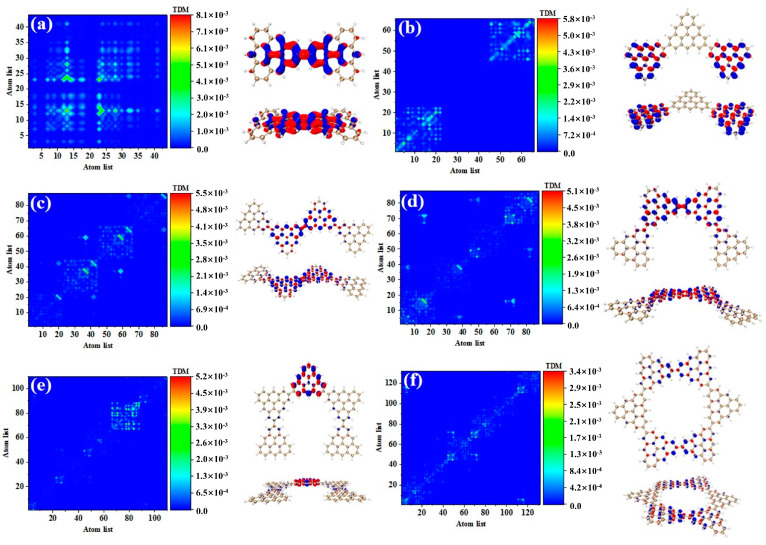
TDM and hole-electron pair density of one-photon excited states of triangulene spin chain N = 2 (**a**), 3 (**b**), 4-1 (**c**), 4-2 (**d**), 5c (**e**), 6c (**f**) TSCs at the main absorption peak.

**Figure 5 molecules-28-03744-f005:**
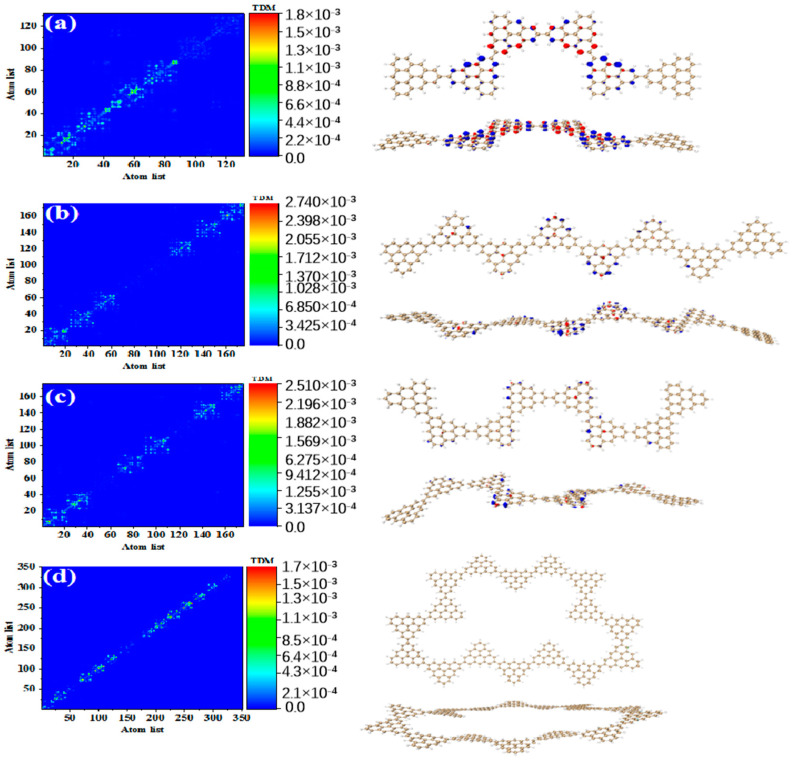
TDM and hole-electron pair density of one-photon excited states of triangulene spin chain N = 6 (**a**), 8-1 (**b**), 8-2 (**c**), 16c (**d**) TSCs at the main absorption peak.

**Figure 6 molecules-28-03744-f006:**
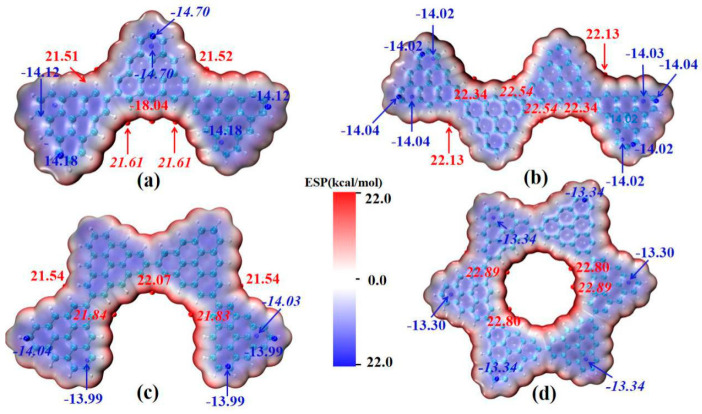
The van der Waals surface electrostatic potential of triangulene spin chain N = 3TSCs (**a**), N = 4-1TSCs (**b**), N = 4-2TSCs (**c**) and N = 6cTSCs (**d**). The red isosurface represents the positive distribution area of ESPO, and the blue isosurface represents the negative distribution area of ESPO, in kcal/mol. The surface local maximum and minimum of ESP are represented by red and blue spheres respectively. Global minimum and maximum values are marked with italic font.

**Figure 7 molecules-28-03744-f007:**
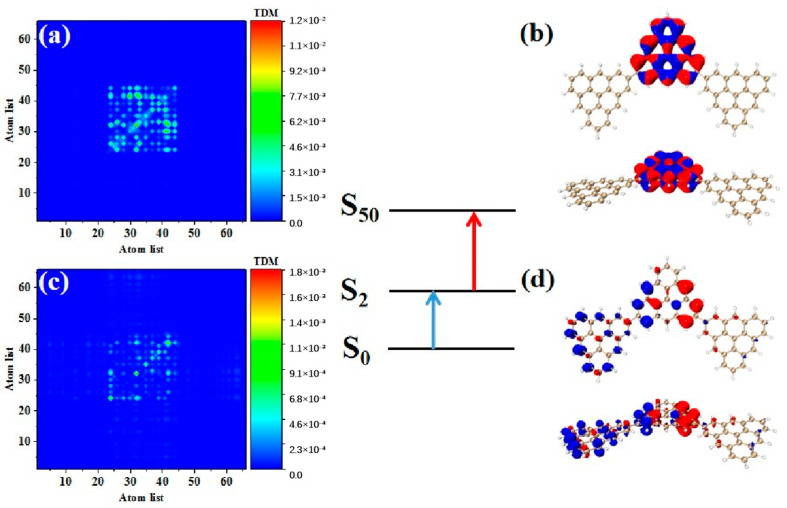
TDM and hole−electron pair density diagram in the first step (**a**,**b**) and second step (**c**,**d**) of TPA at 544.5 nm (S_50_) for N = 3TSCs.

**Figure 8 molecules-28-03744-f008:**
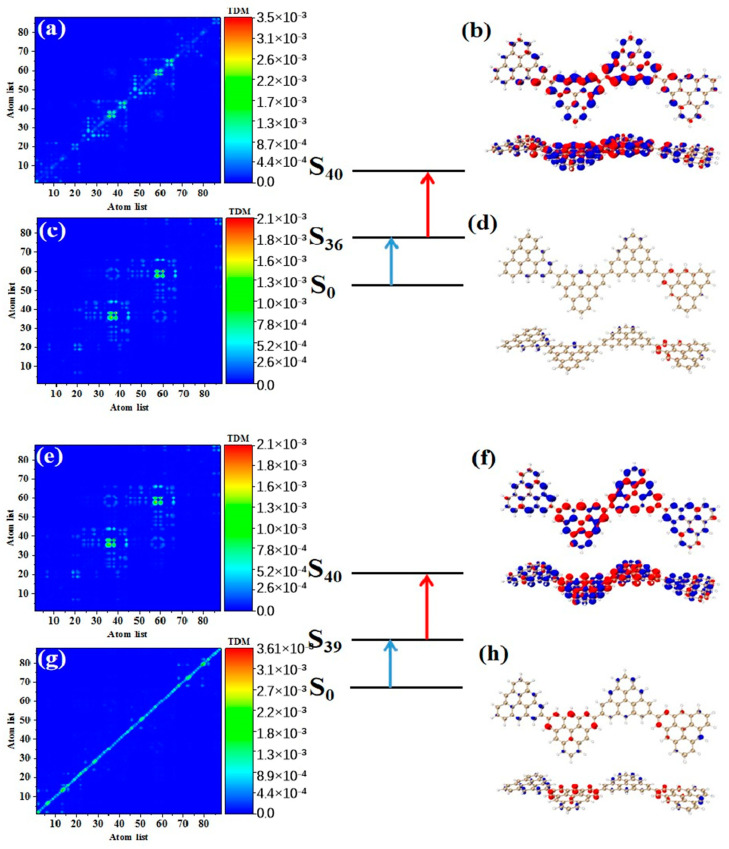
TDM and hole-electron pair density of the first step (**a**,**b**) and second step (**c**,**d**) of TPA when N = 4-1TSCs pass through channel I (intermediate state is S_36_) at 573 nm (S_40_) and the first step (**e**,**f**) and second step (**g**,**h**) of TPA when N = 4-1TSCs pass through channel II (intermediate state is S_39_).

**Figure 9 molecules-28-03744-f009:**
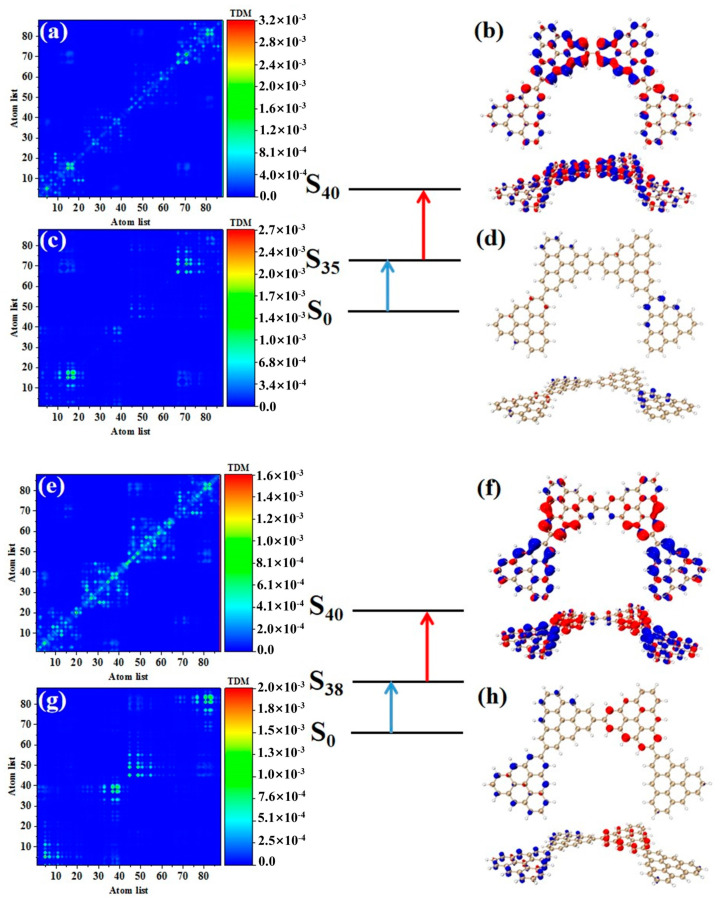
TDM and hole-electron pair density of the first step (**a**,**b**) and second step (**c**,**d**) of TPA when N = 4-2T_S_Cs pass through channel I (intermediate state is S_35_) at 574 nm (S_40_) and the first step (**e**,**f**) and second step (**g**,**h**) of TPA when N = 4-2TSCs pass through channel II (intermediate state is S_38_).

**Figure 10 molecules-28-03744-f010:**
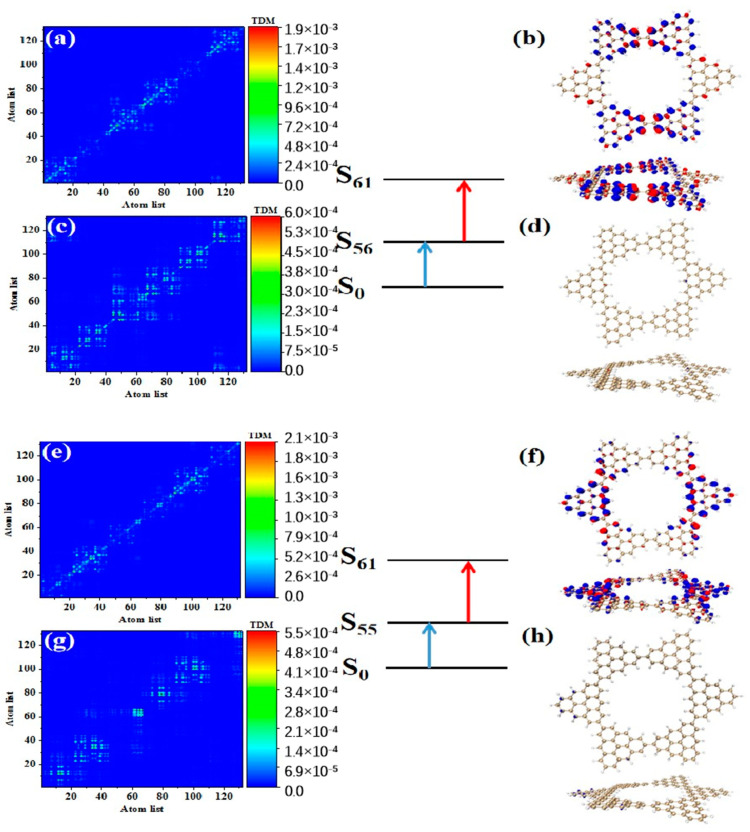
TDM and hole−electron pair density of the first step (**a**,**b**) and second step (**c**,**d**) of TPA when N = 6cTSCs pass through channel I (intermediate state S_56_) at 574 nm (S_61_) and the first step (**e**,**f**) and second step (**g**,**h**) of TPA when N = 6cTSCs pass through channel II (intermediate state S_55_).

**Figure 11 molecules-28-03744-f011:**
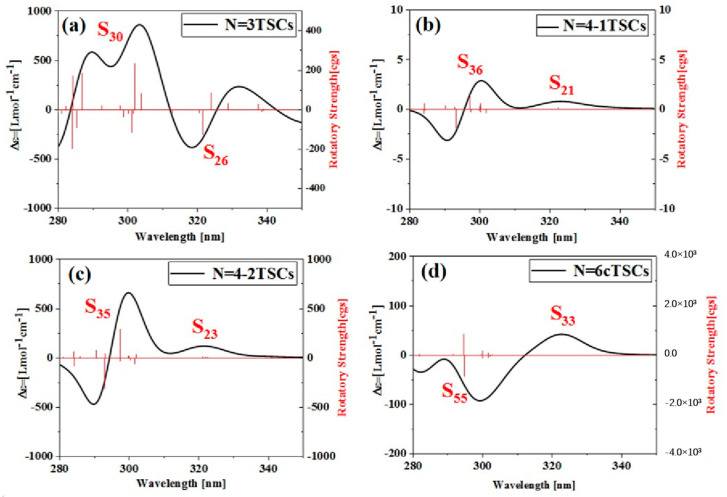
ECD spectra of the spin chain trimer of triangulene ((**a**), N = 3TSCs), the tetramer of triangulene with different connection modes ((**b**), N = 4-1TSCs); ((**c**), N = 4-2TSCs); ((**d**) N = 6cTSCs) and six closed loops of the hexamer of triangulene.

**Figure 12 molecules-28-03744-f012:**
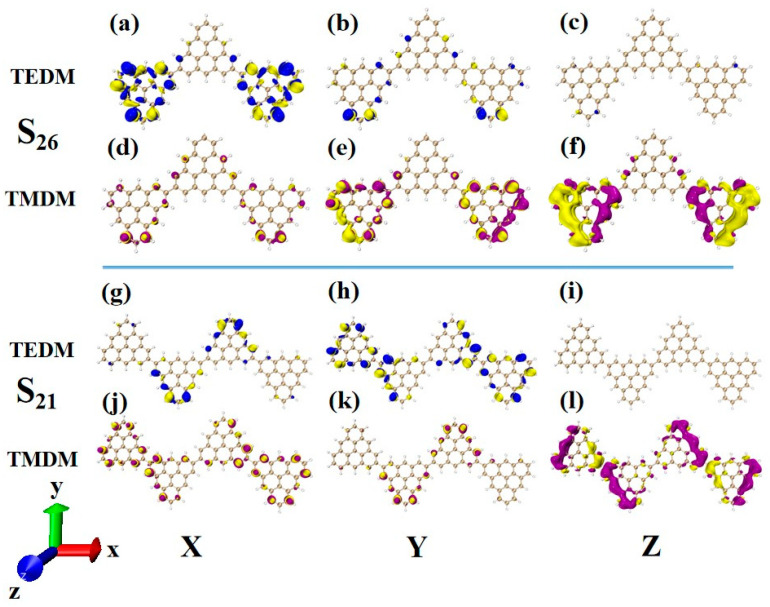
The Cartesian components of TEDM (**a**–**c**) and TMDM (**d**–**f**) of N = 3TSCs in S_26_; N = the Cartesian components of TEDM (**g**–**i**) and TMDM (**j**–**l**) of 4-1TSCs in S_21_ (the blue isosurface in TEDM is positive and the yellow isosurface is negative; the purple isosurface in TMDM is positive and the yellow isosurface is negative).

**Figure 13 molecules-28-03744-f013:**
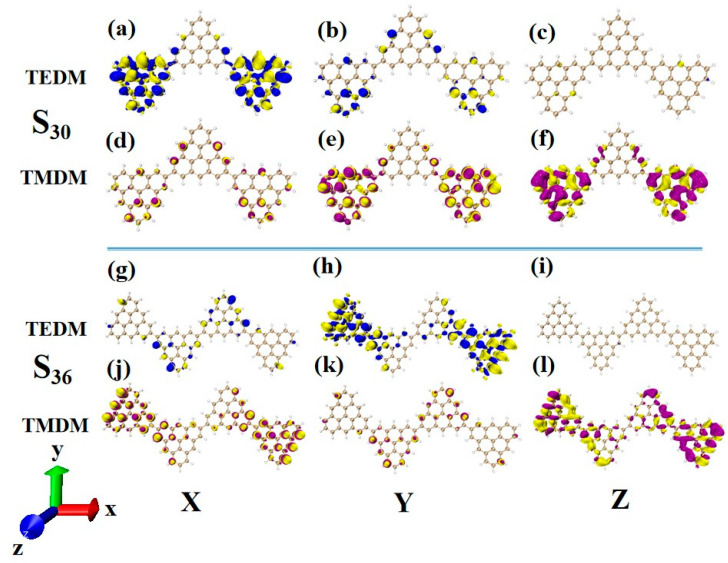
The Cartesian components of TEDM (**a**–**c**) and TMDM (**d**–**f**) of N = 3T_S_Cs in S_30_; N = the Cartesian components of TEDM (**g**–**i**) and TMDM (**j**–**l**) of 4-1TSCs in S_36_ (the blue isosurface in TEDM is positive and the yellow isosurface is negative; the purple isosurface in TMDM is positive and the yellow isosurface is negative).

**Figure 14 molecules-28-03744-f014:**
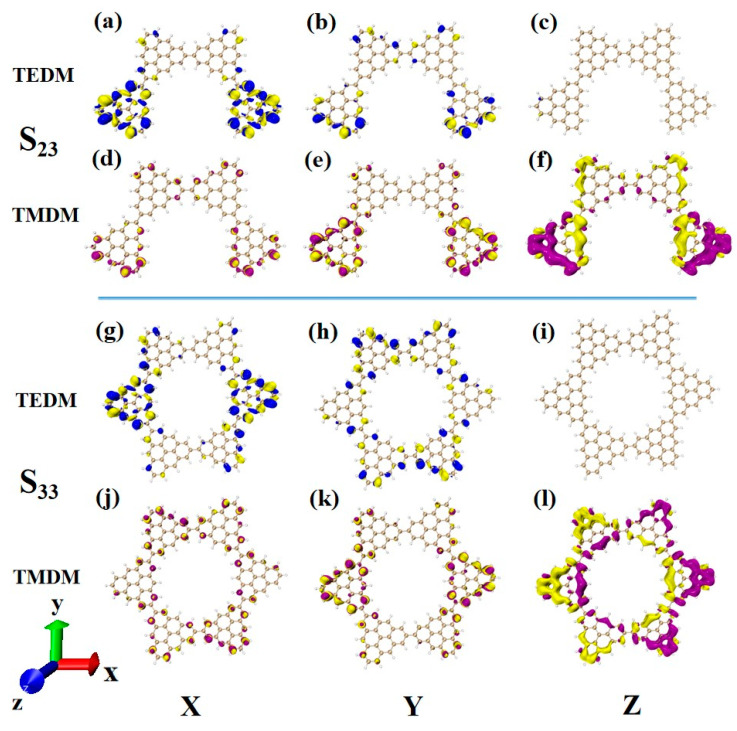
The Cartesian components of TEDM (**a**–**c**) and TMDM (**d**–**f**) of N = 4-2TSCs in S_23_; N = Cartesian components of TEDM (**g**–**i**) and TMDM (**j**–**l**) of 6cTSCs in S_33_.

**Figure 15 molecules-28-03744-f015:**
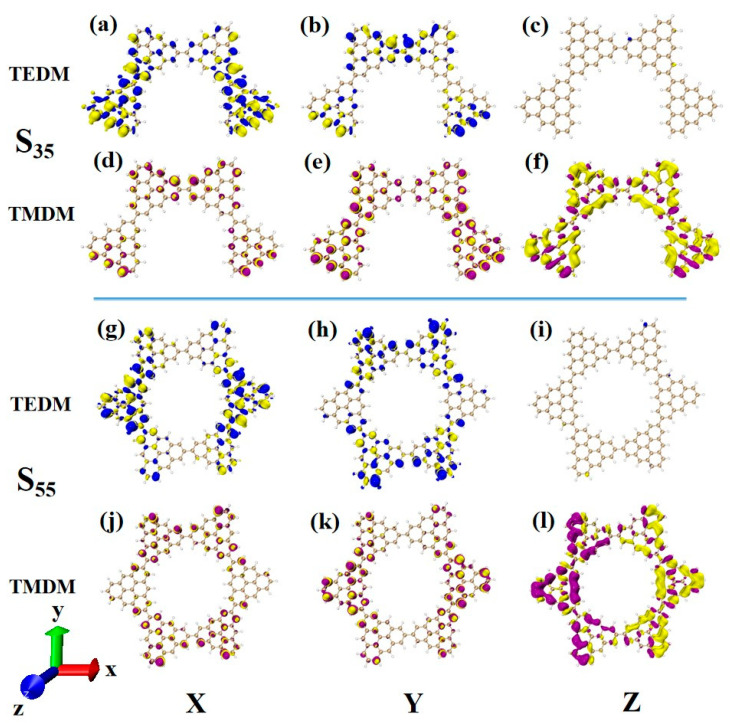
The Cartesian components of TEDM (**a**–**c**) and TMDM (**d**–**f**) of N = 4-2TSCs in S_35_; N = Cartesian components of TEDM (**g**–**i**) and TMDM (**j**–**l**) of 6cTSCs in S_55_.

**Table 1 molecules-28-03744-t001:** One-photon excited state transition index of secondary absorption peak.

	Excited States	Oscillator Strength	Excited Energy (eV)	*H* (Å)	*D* (Å)	*t* (Å)	*S_r_*
N = 3TSCs	S_0_→S_37_	1.3805	4.324	7.322	0.042	−3.535	0.96.64
N = 4-1TSCs	S_0_→S_36_	2.2673	4.175	8.368	0.000	−4.753	0.98551
N = 4-2TSCs	S_0_→S_35_	1.2938	4.173	8.053	0.031	−4.1972	0.98536
N = 6cTSCs	S_0_→S_55_	2.3274	4.210	10.222	0.001	−0.975	0.98753

**Table 2 molecules-28-03744-t002:** One-photon excited state transition index of main absorption peak.

	Excited States	Oscillator Strength	Excited Energy (eV)	*H* (Å)	*D* (Å)	*t* (Å)	*S_r_*
N = 3TSCs	S_0_→S_66_	1.1894	5.037	9.239	0.173	−2.160	0.98102
N = 4-1TSCs	S_0_→S_62_	4.5128	4.748	8.273	0.000	−3.091	0.95292
N = 4-2TSCs	S_0_→S_62_	2.8618	4.744	7.95	0.053	−3.979	0.95280
N = 6cTSCs	S_0_→S_98_	7.0326	4.813	10.111	0.005	−0.960	0.96190

**Table 3 molecules-28-03744-t003:** Elements of transition dipole moment matrix in TPA.

Molecule	State	Path	Process	Integral Value (Debye)	The Largest TPA Cross-Section
N = 3TSCs	S_50_	Ⅰ	$\langle \phi_{s0}\left|\mu \right|\phi_{2}\rangle \times \langle \phi_{s2}\left|\mu \right|\phi_{50}\rangle$	3.96 × 3.96	5.35 × 10^3^
N = 4-1TSCs	S_40_	Ⅰ	$\langle \phi_{s0}\left|\mu \right|\phi_{36}\rangle \times \langle \phi_{s36}\left|\mu \right|\phi_{40}\rangle$	22.39 × 20.59	2.92 × 10^3^
		Ⅱ	$\langle \phi_{s0}\left|\mu \right|\phi_{39}\rangle \times \langle \phi_{s39}\left|\mu \right|\phi_{40}\rangle$	3.25 × 26.27	
N = 4-2TSCs	S_40_	Ⅰ	$\langle \phi_{s0}\left|\mu \right|\phi_{35}\rangle \times \langle \phi_{s35}\left|\mu \right|\phi_{40}\rangle$	12.81 × 15.48	2.19 × 10^3^
		Ⅱ	$\langle \phi_{s0}\left|\mu \right|\phi_{38}\rangle \times \langle \phi_{s38}\left|\mu \right|\phi_{40}\rangle$	10.29 × 17.25	
N = 6cTSCs	S_61_	Ⅰ	$\langle \phi_{s0}\left|\mu \right|\phi_{56}\rangle \times \langle \phi_{s56}\left|\mu \right|\phi_{61}\rangle$	22.32 × 21.15	4.61 × 10^3^
		Ⅱ	$\langle \phi_{s0}\left|\mu \right|\phi_{55}\rangle \times \langle \phi_{s55}\left|\mu \right|\phi_{61}\rangle$	22.98 × 14.18	

## Data Availability

Not applicable.

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
