# Peer review of "Physical Mechanism of Nonlinear Spectra in Triangene"

_molecules, 2023, doi:10.3390/molecules28093744_

Round 1
Reviewer 1 Report
The presented work contains a lot of new data and is undoubtedly interesting for readers. In my opinion, it should be accepted after a minor revision. A few comments are provided below. 1) The TD-DFT theory does not always adequately describe electronic excitations. Different exchange-correlation functionals can give very different results. It would be desirable to validate the method of calculation on some relevant simple (poly)cyclic hydrocarbon whose optical properties are known experimentally. 2) In the caption to tables 1 and 2 the description of the D, H, Sr values would be suitable. Please double-check all units of measurement, they probably have a mistake. 3) Figure 4 is not informative at all. We only see a blue background. It is not clear why red is needed if such values are not achieved anywhere. The figure should be revised. 4) The text is difficult to read due to the large number of abbreviations. A list of abbreviations would be appropriate.
Reviewer 2 Report
Comments:
The manuscript by Weijian Feng, Hanbo Wen, Naixing Feng, Hao Sheng, Zhixiang Huang, and Jingang Wang presents a theoretical study about the linear and non-linear optical absorption properties of open triangulene spin chains and cyclic triangulene spin chains with their length and shape.
Recommendation: The manuscript may be published after major revisions.
- In Tables 1 and 2, it would be convenient to describe the meaning of H, D, t, and Sr. The definition of these concepts is separated from each other in the body of the manuscript.
- Finding a better accommodation for Figure 3 is followed by Figure 2, and according to the wording, it can be inserted later than the current position.
- Correct the name of the software or computer code used for the quantum chemical calculations. The correct term is Gaussian 16.
- PBE1PBE (PBE0) y CAM-B3LYP no son funciones, estos son funcionales.
- Correctly define the functional used for the geometry prediction, the keyword is PBE1PBE, but the name specified in the publication of the functional is PBE0.
- Justify the use of the def2svp basis set.
- It is essential to expose the use of the functional CAM-B3LYP for the calculation in the excited state. The properties exposed in this manuscript depend to a great extent on this aspect.
Reviewer 3 Report
In this manuscript, the authors reported the investigations on linear and non-linear optical absorption properties of open and cyclic triangulene spin chains with different length and shapes. The data are abundant and the conclusions are clear. Therefore it is worth publishing, but the problems in writings should be overcome before acceptance.
1. The writings should be carefully checked. For instance, in Abstract, the full name of TSC is not mentioned, “OPA” should not be abbreviation of “one photon spectra”; in last two paragraph of Introduction section as well as the section 2.1 the “triangulene spin chains (TSCs)” appears repeatedly; In section 2.1, the Figure S1 and S2 are not mentioned with Figure S3 being the first one, similar problem also arises for the sequence of References, which skips from [25] to [37] in section 2.1; In Figure 11, the labels “abcd” are absent;
2. In Abstract, the potential application aspects of such theoretical study were summarized as “manufacture of photoelectric devices, photoelectric functional materials and photochemical sensors based on triangulene”, however, there lacks of specific introduction about these in Introduction section.
3. In Table 1 & 2, the decimal places should be unified to two or three places.
4. The typos or grammatical problems exist, such as, there doesn’t exist Figure 2j as mentioned in Paragraph 2 of section 2.1; the number “0.96.64” in Table 1; the sentence “Calculate dispersion correction energy for geometrically optimized structures using DFT-D3 [30] and exchange-correlation functional” in section 3;
5. In Figure 3 & 4, the names of corresponding secondary absorption peak and main absorption peak could be labeled on the right panels.
6. In Figure 1, only the situation of N=4 and N=8 displays the possibility of different chain configurations, but for N=5 and N=6, the corresponding chain models are neglected. The authors should provide their reason on choosing the models.
7. How about the planarity of the modeled structures, or are they strictly conjugated?
Reviewer 4 Report
The article “Physical mechanism of nonlinear spectra in triangene” is devoted to the influence of the length and shape of the triangulene chains on their one- and two-photon absorption spectra. With the help of quantum-chemical TDDFT calculations and sum-of-states (SOS) theory, the authors considered and discussed in detail the physical mechanism of the local excitation in the triangulene unit and the weak charge transfer between these units. As promoted, it can be helpful in the manufacture of photoelectric devices, photoelectric functional materials, and photochemical sensors based on such materials.
I believe the presented work is very interesting and could be recommended for publication after a major revision. The authors need to address their effort to revise and clarify the next items.
1. English language. There are a lot of mistakes and unclear sentences. For example:
Line 2: I believe it must be “triangulenes” or “triangulene chains” instead of “triangene” in the article name.
Line 93: It is not a clear sentence: “The presence of two unpaired covalent electrons in a single triangulene”.
Lines 95-96: It is not clear, what the authors mean by saying “The unpaired electron … can trap and conduct photoexcited electrons”.
Line 134: “The main absorption peak of the main absorption peak…”.
And so on. These are just a few examples, but authors need to check and revise the whole text.
2. There is a disturbed order of references, figures, and tables quoting. For example, we see Tables 1 and 2 on pages 4 and 5, but the authors quote them for the first time much later, on page 10. It is the same problem in the case of Figures. The authors cite work [37] (line 95) after reference [25] (line 86), and so on.
3. I recommend moving the chapter “Methods” up and placing it after “Introduction”, and moving all theory about calculations of transition density matrix and transition index of the excited state into this chapter. Regarding the DFT method description, the next corrections should be done:
- “Gaussian 16” instead of “Gauss 16”
- “density functional theory” instead of “density general function theory”
- “PBE1PBE functional” instead of “PBE1PBE function”
- “CAM-B3LYP functional” instead of “CAMB3LYP function”
Round 2
Reviewer 2 Report
Accept in present form
Reviewer 4 Report
The authors performed good work, and it seems that article can be accepted now for publication in the revised form. However, there is still one little comment.
The authors use the word “triangene” in the article name, but I guess it must be “triangulene”. Thus, the article's name should be “Physical mechanism of nonlinear spectra in triangulene”. Maybe I’m wrong but check it please again.